# Comparative Investigation of the Physicochemical Properties of Chars Produced by Hydrothermal Carbonization, Pyrolysis, and Microwave-Induced Pyrolysis of Food Waste

**DOI:** 10.3390/polym14040821

**Published:** 2022-02-20

**Authors:** Moonis Ali Khan, Bassim H. Hameed, Masoom Raza Siddiqui, Zeid A. Alothman, Ibrahim H. Alsohaimi

**Affiliations:** 1Chemistry Department, College of Science, King Saud University, Riyadh 11451, Saudi Arabia; mrsiddiqui@ksu.edu.sa (M.R.S.); zaothman@ksu.edu.sa (Z.A.A.); 2Department of Chemical Engineering, College of Engineering, Qatar University, Doha P.O. Box 2713, Qatar; b.hammadi@qu.edu.qa; 3Chemistry Department, College of Science, Jouf University, Sakaka 72388, Saudi Arabia; ehalshaimi@ju.edu.sa

**Keywords:** food waste, char, hydrothermal carbonization, pyrolysis, microwave pyrolysis

## Abstract

This work presents a comparative study of the physicochemical properties of chars derived by three thermochemical pathways, namely: hydrothermal carbonization, HTC (at 180, 200 and 220 °C), pyrolysis, PY, (at 500, 600 and 700 °C) and microwave assisted pyrolysis, MW (at 300, 450 and 600 W). The mass yield of HTC samples showed a decrease (78.7 to 26.7%) as the HTC temperature increased from 180 to 220 °C. A similar decreasing trend in the mass yield was also observed after PY (28.45 to 26.67%) and MW (56.45 to 22.44%) of the food waste mixture from 500 to 700 °C and 300 to 600 W, respectively. The calorific value analysis shows that the best among the chars prepared by three different heating methods may be ranked according to the decreasing value of the heating value as: PY500, MW300, and HTC180. Similarly, a decreasing trend in H/C values was observed as: PY500 (0.887), MW300 (0.306), and HTC180 (0.013). The scanning electron microscope (SEM) analyses revealed that the structure of the three chars was distinct due to the different temperature gradients provided by the thermochemical processes. The results clearly show that the suitable temperature for the HTC and PY of food waste was 180 °C and 500 °C, respectively, while the suitable power for the MW of food waste was 300 W.

## 1. Introduction

Global food wastage continues to grow in alarming proportions, and this is not without far-reaching consequences for the environment [1]. According to the Food and Agricultural Organization, the annual tonnage of wasted food worldwide is around 1.3 billion with roughly 45 to 80% of this waste discarded in landfills [2]. Despite being discarded, this food waste (which is usually processed through energy intensive processes), if properly harnessed, can serve as a valuable resource. With the recent increase in awareness of the contribution of organic waste such as food waste to global warming [3], some countries have begun to craft legislation to ban the disposal of food waste in landfills [4].

Owing to food waste’s rich organic and energy contents, research efforts are being harnessed towards its possible conversion to value-added products such as char instead of being dumped in landfills [5,6]. On a w/w basis, food waste contains ≤33.2 of lipids, ≤29.4% of starch, and ≤23.5% of proteins [7], which makes it have relatively higher hydrogen and, consequently, a higher H/C ratio relative to other biomass types. Although biological conversion methods such as fermentation have been widely used in the promotion of food waste, recent attention has been focused on thermochemical conversion.

Thermochemical processes can be applied for the conversion of municipal food waste to aromatic chars with enhanced calorific content for use as solid fuel in combustion plants [8]. In this study, thermochemical processes such as hydrothermal carbonization (HTC) [9], pyrolysis (PY) [10], and microwave assisted pyrolysis (MW) [11] were employed in the production of char from food waste. During these three processes, the food waste undergoes chemical degradation. While PY involves anaerobic decomposition of waste at 130 to 1300 °C to form a highly dense energy char, HTC converts biomass waste to char via hydrolysis and thermal reactions occurring at temperatures and pressure ranging from 180 to 260 °C and 10 to 20 bar, respectively. On the other hand, the MW of food waste involves the use of microwave energy at power levels between 300 and 2700 W at 10 to 50 bar pressure for food waste pyrolysis. The MW is a useful method for shortening process time, saving energy, and appreciating the quality of the char.

Some important physicochemical properties of char include oxygen-groups’ surface functionality and thermal stability, which makes it suitable for the adsorptive removal of metal ions present in wastewater [12]. Another physicochemical property of food waste-derived char is the high carbon content (45–93%), high energy values (3585–7170 kcal/kg), and low ash content (2.3–6.4%), which qualifies it as a substitute for solid fuel [13,14]. The fuel properties of a char are considered to vary depending on the thermochemical method and/or the mechanism involved [15].

The HTC process in particular is a conventional but recently revived thermochemical process [16] adopted by researchers for the valorization of globally generated food waste. Temperature and time are the most important parameter for HTC, while temperature and power are important operating parameters in the PY and MW of food waste since they have a significant effect on the final products. Moreover, MW is characterized by a volumetric heating mechanism which is different from the conduction, convection, and radiation heating modes of conventional PY and HTC [17]. These distinctive heating mechanisms for the three thermochemical processes are expected to bring diverse change in the physical and chemical properties of the final products.

Hence, more comparative studies conducted under different process conditions are needed to validate the efficacy of the HTC process among other thermochemical processes for food waste conversion to char. Comparative studies to investigate the physicochemical properties of food waste-derived chars produced by HTC and PY were carried out [17,18]. However, the mass yield and heating values of the char produced during the study were not analyzed. These properties would provide useful insight into their suitability as solid fuel. The mass yield and heating value of hydrochars and pyrochars were analyzed by Theppitak and co-workers [19]. However, to the best of our knowledge, no single work has been conducted on the production of char using three different reaction conditions. Thus, the objective of the current study was to develop hydrochar, pyrochar, and microchar from the food waste mixture through HTC, PY, and MW, respectively. The second objective was to characterize the produced char and assess its suitability as solid fuel.

## 2. Experimental

### 2.1. Preparation of Food Waste Mixture

A feedstock sample comprised of 15% dates (without stones), 20% lentils, 20% chickpeas, 20% rice, and 25% orange peel by weight % was prepared by mixing the constituent food waste and drying in an oven for 24 h. Three (3) portions each of 10 g of the mixed samples (1 mm size) were weighed and kept in a desiccator in a plastic bag in preparation for further experiments. Each sample was thermo-chemically converted to char via the HTC, PY, and MW routes.

### 2.2. Preparation of Char by Hydrothermal Carbonization

The first portion of the sample was thoroughly mixed with 90 mL of deionized (D.I) water (sample: water mass ratio = 10% *w*/*w*) and transferred into an automated stainless-steel hydrothermal reactor (170 mL). After prior stirring for 30 min, the temperature increased from room temperature at a heating rate of 5 °C/min until it reached 180 °C and was held at this temperature for 120 min. At the end of the set reaction time, the reactor was cooled slowly to room temperature. The hydrochar particles were subsequently separated from the reaction mixture by vacuum filtration and repeatedly washed with hot distilled water. The samples were dried in an oven (60 °C) for 24 h. The hydrothermally carbonized sample was kept in a small plastic sample bag and code named HTC180. This process was repeated at 200 and 220 °C to obtain HTC200 and HTC220, respectively.

### 2.3. Preparation of Char by Conventional Pyrolysis

The second portion of the sample was placed in a tubular GSL-1100X quartz furnace (0.025 ≈ 0.051 m diameter) equipped with a programmable temperature controller. The N_2_ gas (99.99%) was simultaneously supplied to the furnace at a 200 cm^3^/min flow rate. The temperature of the furnace was raised from an ambient temperature to 500 °C during first experimental run. The sample remained at this temperature for 90 min inside the furnace and was thereafter rapidly cooled to an ambient temperature under N_2_ gas flow. The pyrolyzed char that was produced was kept in a small plastic sample bag and code named PY500. The second and third experimental runs were also conducted at 600 and 700 °C and the sample chars produced were tagged PY600 and PY700, respectively.

### 2.4. Preparation of Char by Microwave Induced Pyrolysis

The third portion of the sample was placed inside a quartz cell and transferred into a modified microwave oven (Samsung ME711K, 20 L). The sample was irradiated with microwave energy at 300 W for 20 min under a 100 cm^3^/min flow of pure N_2_ (99.99%) gas. The microwave-pyrolyzed sample was allowed to cool inside the reactor under N_2_ gas flow to room temperature and packaged in a small plastic sample bag with the tag MW300 assigned to it. The second and third experimental runs were conducted in the same manner, but the microwave irradiation power was varied to 450 and 600 W to obtain MW450 and MW600 samples, respectively.

### 2.5. Physical and Chemical Property Analysis of the Chars

#### 2.5.1. Mass Yield Analysis of the Chars

The mass yields of the char samples were calculated using the formula:(1)$$Mass\;yield_{char}=\frac{m_{char}}{m_{raw}}\times 100$$
where $Mass\;yield_{char}$ is the yield of char in %, $m_{char}$ is the mass of char in g, and $m_{raw}$ is the mass of raw food waste mixture in g.

#### 2.5.2. Elemental Analysis

The elemental analysis was conducted by placing 0.002–0.004 g of sample powder in an Elementar—CHNS analyzer (VARIO EL III) supplied by Analysen systeme GmbH. The percentages of carbon (C), hydrogen (H), nitrogen (N), oxygen (O), and sulphur (S) present in the samples were then determined. The O was calculated by difference.

#### 2.5.3. Calorific Value Analysis

The calorific values of the samples (0.05–0.1 g) were determined using a bomb calorimeter (IKA-WERKE C-5000 model).

#### 2.5.4. Fourier Transform Infra-Red Analysis

In order to track the changes of the char surface functional groups of the three samples after conversion of the food waste, an FT-IR test was conducted on the food waste and chars using an FTIR spectrometer (Perkin-Elmer, Spectrum RXI, Norwalk, CT, USA). The samples were first mixed thoroughly with KBr in a ratio of 1:100 and then compressed into solid thin discs, and the absorption data obtained in the spectrum range 4000–400 cm^−1^ was recorded.

#### 2.5.5. Scanning Electron Microscopic Analysis

A scanning electron microscope (Hitachi, TM3030Plus, Tabletop Microscope, Tokyo, Japan) was used to obtain the SEM images at 10,000× magnification. The samples were prepared for the electron scanning microscopy by sputter coating the samples with gold to avoid the charging effect and to obtain a good-quality SEM image.

## 3. Results and Discussion

Figure 1 illustrates the mass yield of the chars produced by the HTC, PY, and MW of the food waste mixture. The mass yield of the HTC samples showed a decrease (78.7 to 26.7%) as the HTC temperature increased from 180 to 220 °C. Wei et al. [20] explained that the drop in HTC yield may be a consequence of intense water loss and removal of carboxylic acid at elevated temperatures. A similar decreasing trend in the mass yield was also observed after PY (28.45–26.67%) and MW (56.45 to 22.44%) of the food waste mixture from 500 to 700 °C and 300 to 600 W, respectively. This sequential decline in the mass yield was also reported by Liu and colleagues [21] for MW heating of real food waste from 300 to 600 W, with a maximum yield of 81.63% obtained during their study.

Thus, the HTC of food waste at 180 °C, PY of food waste at 500 °C, and MW of food waste at 300 W gave the optimum mass yield of the chars. It is also clear from Figure 1 that the yield of HTC180 was higher than that of PY500 and MW300, which is due to the different mechanisms of the carbonization processes. For instance, the yield of the char produced via HTC at 180 °C was higher than the yield obtained via PY and MW at 500 °C and 300 W, respectively.

The high mass yield of hydrochar that was obtained in this study could be due to initial hydrolysis of the lentil and chickpea components of the food waste and subsequent reactions that formed a completely new structural material [22]. In the case of PY and MW, due to the elevated temperatures used, higher mass losses were recorded due to the high-temperature sensitivity of lentils and chickpeas, which together make up 40% of the food waste mixture.

These samples were examined further in terms of elemental and higher heating value (HHV) analysis and the results are presented in Table 1. From Table 1, it can be clearly seen that the percentage of elemental C content of the pyrochar was the highest (72.73%). This value of elemental C for the PY, as well as the value for the hydrochar (48.23%) and microchar (54.02%) obtained during the current study, were very low compared to the microchar yield of 90% obtained from similar simulated food waste [11]. This may be attributed to the high vegetable component (45%) used in the food waste mixture. It is posited, according to the literature, that the high C content could be due to the preponderance of aromatization reactions taking place during pyrolysis processes.

It can also be seen in Table 1 that, with the exception of the HTC process where there was an increase in elemental H (4.75 to 5.21%), a decrease of 4.75 to 0.94% and 4.75 to 3.98% for H was observed during the PY and MW of the food waste mixture, respectively. Given the energy intensive nature, a decrease in H during the PY and MW processes could be explained by the fact that a dehydration reaction through loss of hydroxyl (–OH) groups from the food waste mixture could have occurred. It is also clear from Table 1 that PY500 produced through pyrolysis at 500 °C had the highest loss of H with the final value reaching 0.94% from 4.75% at the end of the reaction. Hu and Gholizadeh [23] asserted that the predominant chemical reactions occurring at pyrolysis temperatures below 500 °C are dehydration, decarboxylation, and decarbonylation, while dehydrogenation is the major reaction that takes place above 500 °C. These reactions could be responsible for the higher loss of H present in the chars after PY.

It was further observed that after HTC and PY of the food waste mixture, there was a slight increase in elemental N to 4.01 and 3.94%, respectively, from 3.89% originally found in the food waste mixture. The enrichment of the N, especially during hydrochar production, may be due to the intensive hydrolysis of the protein-rich food waste mixture to nucleic acids [24]. In the literature, it has recently been pointed out that the relatively high percentage of N in protein-rich food waste-derived hydrochar has continued to be a major challenge hindering its commercialization [25] or possible use as solid fuel. For MW of the food waste mixture, elemental N decreased slightly from 3.89 to 3.84%. The slight reduction could be caused by the splitting **of** amino acids, proteins, nucleotides, and loss of gaseous N products [26].

The variation of elemental O after HTC, PY, and MW is also presented in Table 1. A careful look at the results clearly indicates an expected loss of O during all the three processes. The PY (PY500) had the highest O loss (54.2%). The significant loss in O is noteworthy because it is a suitable replacement for solid fuels. During thermochemical conversion of food waste, sizeable numbers of oxygenated compounds could be lost by deoxygenation and dehydration reactions.

Presented in Table 1 is the percentage S content of the food waste mixture and the derived chars. The percentage S content (0.54%) in the food waste mixture increased to 0.74% after HTC. It may be inferred that the HTC increased the quantity of elemental S content of the food waste mixture. However, the major shortcoming is the generation of acidic sulfur oxide during the combustion of the char, which may be detrimental to the environment.

The H/C and O/C ratios of the food waste mixture and the three different chars are presented in Table 1. It is clear from Table 1 that the PY had a relatively low H/C molar ratio compared to the HTC and MW, which may be interpreted to indicate a high level of coalification and aromatic contents. As shown in Table 1, the O/C and H/C ratios for the food waste mixture were 1.19 and 0.114, respectively. The O/C ratio was higher than 0.548 and the H/C ratio was lower than 1.702 reported for oil-extracted food waste [27]. The difference in the O/C and H/C ratios in the work under reference and the current study may be attributed to the difference in the chemical composition of the feedstocks.

After carbonization by PY, HTC, and MW, there was a general reduction in the O/C and H/C ratios, which was evidence of depressed energy content. For instance, the O/C and H/C ratios for the HTC180 reduced to 0.867 and 0.108, respectively. This indicated the transformation into hydrochar owing to dehydration, decarboxylation, and demethanation reactions. This result is within/outside the range reported for hydrothermal carbonization of oil-extracted food waste, where the O/C and H/C ratios were 0.307–0.111 and 1.426–1.573, respectively [26].

For PY500, the O/C and H/C ratios were reduced to 0.3059 and 0.0129, respectively, which is an indication of the scission of the H and the O atoms (by dehydration and decarboxylation) from the functional groups of the food waste mixture, which is in line with previous research [28]. The O/C and H/C ratios for the MW300 reduced to 0.698 and 0.0737, respectively. This is also attributable to a dehydration reaction. In another related study involving food sludge, the H/C and O/C atomic ratios of microchar were reported to be 1.92 and 0.46, respectively [29].

The fuel quality of a char largely depends on its heating value [30]. The heating values of the food waste mixture and the highest yielding chars obtained from each of the thermochemical routes are presented in Table 1. It is quite evident from Table 1 that the HHV of PY500 (5864 kcal/kg) was not only higher than 3851 kcal/kg heating value obtained for the food waste mixture but also higher than 4517 and 5023 kcal/kg heating values obtained for HTC180 and MW300, respectively. Although the yield of HTC180 was higher than both PY500 and MW300 indicating that the HTC process could recover more char, the heating value of the PY500 was higher, indicating that pyrolysis conversion could retain most of its energy.

Figure 2 presents the FT-IR spectra results of the food waste mixture, HTC180, PY500, and MW300 samples. The result shows quite a number of shoulder peaks and few sharp peaks. The vibration around 3840 cm^−1^ in the food waste mixture spectrum may be assigned to the O–H functional group. With the exception of the HTC spectrum, this vibration was also found in the PY and MW spectra. A peak found at 2931 cm^−1^ on the food waste mixture spectrum could be attributed to stretching of the C–H bonds of CH_2_ found in the orange peel component of food waste. This also implies that the food waste mixture is aliphatic in nature. The shoulder peak at 2350 cm^−1^ signifies small levels of absorption of CO_2_ on the surface of the food waste mixture. The sharp peak around 1649 cm^−1^ indicates vibration associated with the combined bending of the N–H bonds and stretching of the C–N bonds, which is usually found in amide complexes. This is indicative of the fact that protein-rich chickpeas and lentils are among the food components of the food waste mixture. The presence of a carbohydrate (rice) component in the food waste mixture was confirmed by the appearance of C–O–C absorption peak at 1032 cm^−1^. The absorption band at 799 cm^−1^ corresponds to the vibration of C–H bonds found in aromatic compounds.

The HTC180, PY500, and MW300 FTIR spectra illustrated in Figure 2 for comparison also displayed a number of shoulder peaks but at absorption frequencies slightly different from the food waste mixture. The difference in peak frequency may be due to different reaction mechanisms involved [31] in each of the carbonization processes. It is clearly shown in Figure 2 that the PY500 FT-IR spectrum, especially between 2000 and 3500 cm^−1^, was flat, which indicated that the pyrolysis thermochemical route might have decomposed most of the lipid-containing orange peel component of the food waste mixture. The appearance of three peaks at the far-flung absorption frequencies at 1416, 1044, and 810 cm^−1^ were strong evidence of the C–O skeletal stretch of the aromatic compound, O–H vibration/C–O stretch, and C–H tri-substituted compounds, respectively.

The spectrum of the HTC180 sample also presented in Figure 2 shows intensive bands around 1030 cm^−1^, which is attributed to the Si–O stretching vibrations, indicating the presence of silica. A weak peak at 800 cm ^−1^ was assigned to the Si–O stretch-related vibration, indicating the formation of an Si–O–Si skeletal network within the char after HTC reaction. A weak peak at 1424 cm^−1^ hints at the presence of phenol due to its O–H bending/C–O stretching structural vibration, while the relatively intense band at 1651 cm^−1^ was associated with the vibration of the O–H bond. The aromatic stretching near 1528 cm ^−1^ could be a result of the presence of nucleic acids on the surface of the hydrochar.

The MW300 FT-IR spectrum followed a wave pattern similar to the food waste mixture feedstock. The MW300 FT-IR spectra showed an absorption band at 795 cm^−1^ associated with deformation of the C–H bonds in aromatic compounds. A sharp and intense peak at 1037 cm^−1^ was associated with C–O–C and C–OH deformation. Compared to the food waste mixture, MW300 and HTC180 showed a strong absorbance at 1037 cm^−1^. Furthermore, the fingerprint region of MW300, which was close to 1415–1653 cm^−1^, had peaks due to the COO– and C–O stretches.

Figure 3 presents the scanning electron micrograph (SEM) images of HTC180, PY500, and MW300 as well as the food waste mixture. Figure 3a shows the original structure of the food waste with irregular and large clumps of particles on its surface. However, the microstructure of the three chars followed according to the carbonization degree [4] of the (cellulose, hemicellulose, and lignin) components present in the food waste and the temperature gradient provided during the thermochemical process [32]. During HTC, the cellulose and hemicellulose components of the food waste are converted into char, while during PY and MW all three components are carbonized. As illustrated in Figure 3b, the hydrochar shows a smoother surface with some cracks and crevices. The sparse carbon spheres observed for HTC arises from the simultaneous hydrolytic and thermal conversion carbohydrates into secondary char (spheres) that leaves behind a skeletal framework of the lignin structure. The surface of the PY (Figure 3c) appears to be rougher, looser, and more porous than that of the obtained HC. This is evidently due to the evolution of gases (e.g., CO, CO_2_, H_2_, CH_4_, and H_2_O) generated by the decomposition of food waste at a relatively higher temperature. It was observed that the surface of the PY is covered with both uniform and non-uniform sized regular particles, probably as a result of the surface roughness produced by the pores. Moreover, pyrolysis provides a temperature gradient from the hot surface to the interior of each char particle, and this may explain the observed heterogenous microstructure of the PY. The microstructure of the MW300 surface (Figure 3d) produced by microwave-aided pyrolysis is rare and the only one of its kind. The microchar structure appeared nano-flower-like or as starch flakes. This may be due to the dehydration and breaking of the bonds of the organic complex formed as a result of the excitation of food waste molecules caused by microwave energy. Unlike the HTC and PY, the temperature gradient in the MW occurs in reverse order from the internal core of the MW to the lower temperature regions at the surface of the char.

Figure 4 displayed the physical attributes of the food waste mixture, HTC180, PY500, and MW300. The food waste mixture appeared brownish-orange in color, probably because it had a large quantity of lentils and chickpeas. After HTC of the food waste mixture, the resulting hydrochar residue at 180 °C had a reddish brown/seal brown color, different from the black color of the PY and MW residue obtained at 500 °C and 300 W, respectively. The reddish-brown color of hydrochar shows the extent of graphitization of the food waste mixture, while the black color of both PY500 and MW300 have an appearance close to natural coal.

## 4. Conclusions

The result of this study shows that PY, MW, and HTC under selected and specified carbonization conditions could be used to convert food waste mixtures into chars for specific applications such as fuels. A comparison of the physicochemical properties of chars produced by HTC, PY, and MW of a food waste mixture under different selected reaction conditions shows that the yield of hydrochar was higher than both pyrochar and microchar. Among the produced chars, the calorific value (5864 kcal/kg) and oxygen loss (54.2%) was highest for PY. This result was also consistent with the respective H/C and O/C ratios. This study further revealed that the PY had the potential to produce char with HHV comparable to coal.

## Figures and Tables

**Figure 1 polymers-14-00821-f001:**
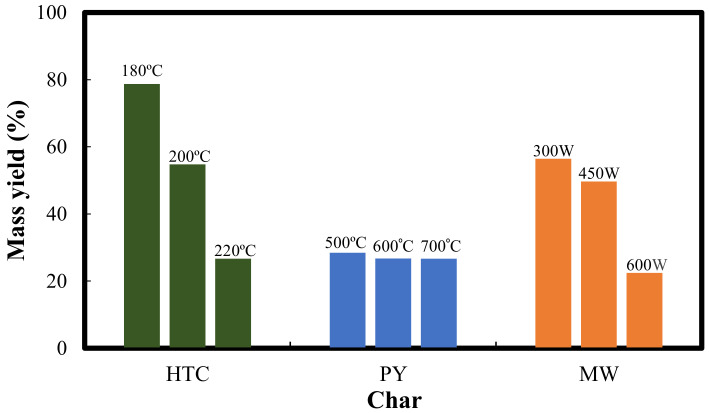
Mass yield of char produced by hydrothermal carbonization, pyrolysis, and microwave induced pyrolysis.

**Figure 2 polymers-14-00821-f002:**
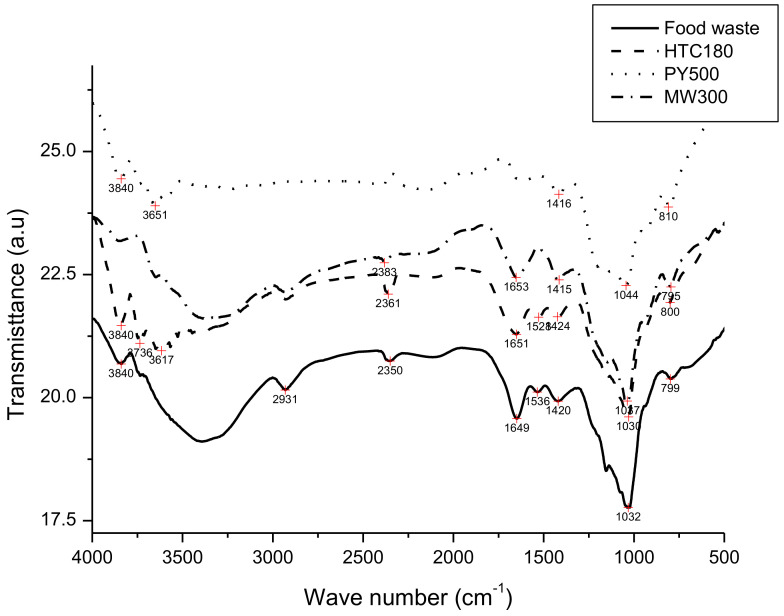
FT-IR spectra of food wastes mixture, HTC180, PY500, and MW300.

**Figure 3 polymers-14-00821-f003:**
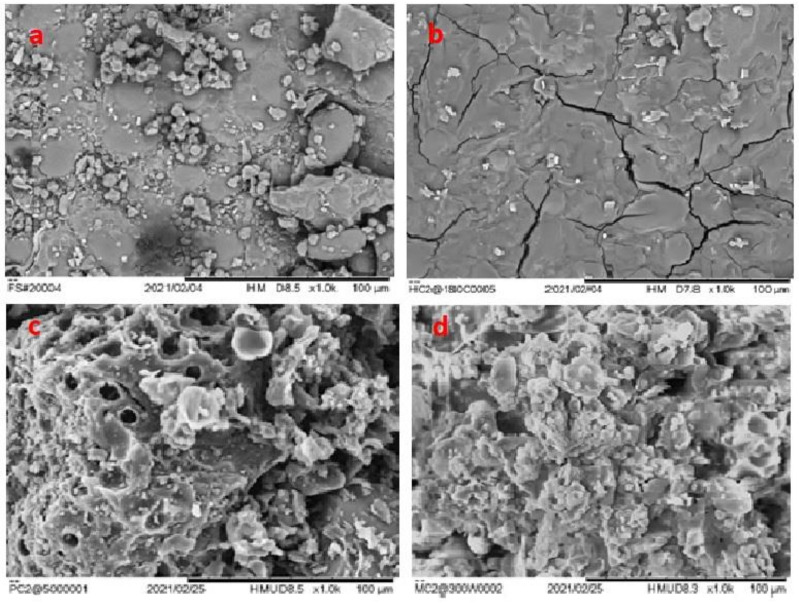
SEM images for (**a**) Feedstock, (**b**) HTC180, (**c**) PY500, and (**d**) MW300 at 1000× magnification power.

**Figure 4 polymers-14-00821-f004:**
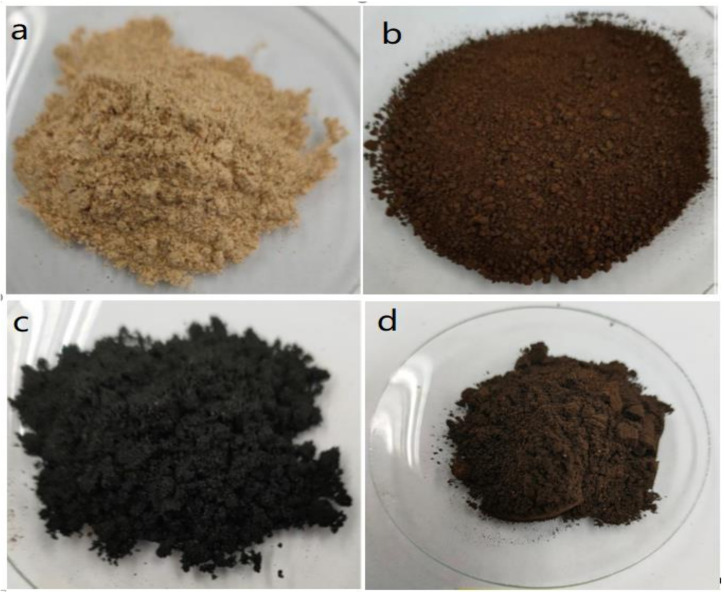
Photographic images for (**a**) Feedstock, (**b**) HTC180, (**c**) PY500, and (**d**) MW300.

**Table 1 polymers-14-00821-t001:** Elemental and calorific values of food wastes mixture and chars with highest yield.

Parameter	Food Wastes	HTC180	PY500	MW300
Elemental Analyses				
C (%)	41.44	48.23	72.73	54.02
H (%)	4.75	5.21	0.94	3.98
N (%)	3.89	4.01	3.94	3.84
S (%)	0.54	0.74	0.14	0.46
O (%)	49.38	41.81	22.25	37.70
O/C	1.19	0.87	0.31	0.70
H/C	0.11	0.11	0.01	0.07
Calorific Value				
HHV (kcal/kg)	3851.00	4517.00	5864.00	5023.00
LHV (kcal/kg)	3751.00	4459.00	5769.00	4952.00
Ash content (%)	4.07	6.19	6.27	7.45

## Data Availability

The data presented in this study are available on request from the corresponding author.

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
