# Peer review of "Comparative Investigation of the Physicochemical Properties of Chars Produced by Hydrothermal Carbonization, Pyrolysis, and Microwave-Induced Pyrolysis of Food Waste"

_polymers, 2022, doi:10.3390/polym14040821_

Round 1

Reviewer 1 Report

In this report, the Author mentioned, “Comparative investigation on physicochemical properties of chars produced by hydrothermal carbonization, pyrolysis, and microwave-induced pyrolysis of food waste”. The reported work is very good for readers, so recommended to accept after major comments.  

  1.  
  2. “In addition, the respective H/C and O/C ratios for HTC180, PY500, and MW300 were found to be 0.108 and 0.887, 0.013 and 0.306, and 0.074 and 0.698”, in this section author should mention units.
  3. Authors should revise the abstract to a more scientific perspective.
  4. Authors should revise the Introduction section in more detail in the current study.
  5. The quality of Figure 2 improved in the current format of the Journal, peak position add in Figure.
  6. Figure caption of SEM not clear, So the author should correct caption and color a, b, c, and d. also author provided different magnification images of all samples.
  7. Author mentioned in the section, 2.5.2. Elemental analysis, but the author should provide more details of Elemental analysis like EDS, XPS etc.
  8. Author should explain microwave properties in details.

Reviewer 2 Report

Manuscript Number: polymers-1590898

Full Title: Comparative investigation on physicochemical properties of chars produced by hydrothermal carbonization, pyrolysis, and microwave-induced pyrolysis of food waste

Article Type: Original Research

Dear Sir,

The authors reported “Comparative investigation on physicochemical properties of chars produced by hydrothermal carbonization, pyrolysis, and microwave-induced pyrolysis of food waste”. It is quite good work; therefore, I feel that this manuscript can be published in the esteemed journal ‘Molecules’ after the Minor Revision, following few Comments:

Comments-

  1. Authors can explain the abstract part in briefly.
  2. Please enhance the image quality of the graph of FTIR and in SEM images.
  3. Please rewrite the analysis of SEM image more scientifically according to the synthetic methods.
  4. Please use this ref. in Introduction and FTIR section (1) ChemistrySelect. 4 (2019) 6233–6244, (2) Int. J. Hydrogen Energy. 45 (2020) 18623–18634, (3) Renew. Sustain. Energy Rev. 154 (2022) 111877 and (4) J. Hazard. 421 (2022) 126741.
  5. Please check all the references in Ref sec, it is not uniformly maintained and some where it is in different language. Please have a look.

Round 2

Reviewer 1 Report

I agree with the changes made and believe the manuscript can be submitted for publication in present form; the only minor comment on the Figure 1- please check  Figure 1 in the revised and old versions.